# Nanoparticle-Mediated Signaling for Aptamer-Based Multiplexed Detection of Cortisol and Neuropeptide Y in Serum

**Naimesh Sardesai [1], Yi Liu [1], Regina Shia [2], Peter Mirau [2], Jorge L. Chávez [3] and Nathan S. Swami [1,*]**

[1] Department of Electrical & Computer Engineering, University of Virginia, Charlottesville, VA 22903, USA; sardesai.chem@gmail.com (N.S.); yl2tu@virginia.edu (Y.L.)

[2] Materials and Manufacturing Directorate, Air Force Research Laboratory, Wright-Patterson Air Force Base, Dayton, OH 45433, USA; regina.shia@us.af.mil (R.S.); peter.mirau@afresearchlab.com (P.M.)

[3] 711th Human Performance Wing, Air Force Research Laboratory, Wright-Patterson Air Force Base, Dayton, OH 45433, USA; jorge.chavez_benavides.2@us.af.mil

* Correspondence: nswami@virginia.edu

**Abstract:** Multiplexed profiling of the expression of neurochemical biomarkers of stress, for periodic assessment to enable augmentation of human performance, requires wash-free detection platforms that exhibit reproducible signals from samples in biological matrices. However, alterations in aptamer conformation after binding to targets, such as cortisol, are minimal based on NMR spectra, and the methylene blue signaling is blocked by serum proteins. Hence, in this study, we explore aptamer derivatization with magnetic nanoparticles that are conjugated with multiple methylene blue moieties, to amplify signals and alter the net charge configuration for repulsing serum proteins, so that the aptamer conformation upon target recognition can lead to a signal ON assay in serum media. Based on this, a microchip platform with addressable electrodes that are immobilized with selective aptamer receptors is developed for multiplexed detection of cortisol (1–700 ng/mL) and neuropeptide Y (5–1000 pg/mL) in patient-derived serum samples, which is validated by immunoassays. We envision the application of this sensor for profiling a wider array of human performance biomarkers under stress-related events to develop stress augmentation methodologies.

**Keywords:** stress; voltammetry; aptamer; nanoparticle; microfluidics; NMR

## 1. Introduction

Stress conditions due to physical, environmental, and psychosocial factors cause alterations in the dynamic state of equilibrium or homeostasis within the brain [1,2], eliciting a stress response that includes biochemical, physiological, and behavioral modifications to peripheral tissues [3]. This response can be characterized based on the level and temporal expression profile of small molecules, proteins, nucleic acids, and lipids in peripheral fluids, such as blood, plasma, saliva, interstitial fluid, and sweat [4]. There is great interest in monitoring the expression profiles of these neurochemical biomarkers under relevant stress perturbations [1,2] for the assessment of human performance so that procedures for regulating stress can be developed [5].

Biomarkers of interest include neuropeptide Y (NPY), which is released from sympathetic neurons to alleviate stress due to its mitogenic and pleiotropic effects on peripheral cells [6], glucocorticoids including free cortisol, which regulates stress-related physiological processes [7], and dehydroepiandrosterone sulfate (DHEAS), which activates neurotrophins to cause androgenic and estrogenic hormonal alterations that regulate stress [8]. However, measurement of these macromolecular expression profiles to characterize stress, vigilance, and anxiety is particularly challenging, due to the low levels at which key human performance biomarkers are present within peripheral fluids versus a matrix of interfering proteins and the extremely low sample volumes available for these measurements. To

correlate short-term stress cycles with long-term stress resilience, we seek to simultaneously quantify the levels of cortisol, which exhibit periodic stress-induced variations, as well as NPY levels, which provide information on stress resilience.

Recent reports [9–12] have utilized aptamer-based assays to detect either cortisol or NPY in biological samples of interest, but multiplexed detection of both biomarkers in relevant biological fluids and their validation against alternate detection methods in human samples has not been reported. Additionally, alterations in aptamer conformation upon target binding were not studied, which can lead to protocols for optimizing target binding in the presence of serum proteins. In this study, we use the insights from NMR spectroscopy on aptamer conformation to explore nanoparticle-mediated signaling for sensor platform development and optimize its response to the targets.

Herein, we report a multitarget sensor platform for cortisol and NPY on a microchip platform, using highly selective aptamer receptors for target recognition and validate this platform by comparison with immunoassay results for the two analytes using serum samples from human samples. The innovative aspects of the presented sensor, based on the advancements over our prior research [13–17], include the development of (1) nanoparticle-mediated methylene blue (MB) signaling to create signal ON sensors under aptamer conformation alteration upon binding to the target of interest; (2) signal amplification by attachment of multiple MB moieties to each nanoparticle to enhance signaling from specific versus non-specific binding events; (3) multiplexed electrochemical sensing of the two targets (cortisol and NPY) on a microchip platform that requires sub-μL sample volumes. We envision the application of this sensor in a wearable format [18], to extract markers from interstitial fluid for profiling a wider array of human performance biomarkers from biological matrices [19], under stress-related events to develop assessment and augmentation methodologies.

## 2. Materials and Methods

### 2.1. Materials and Reagents

Thiolated cortisol aptamer (5′-/5AmMC6T/GG AAT GGA TCC ACA TCC ATG GAT GGG CAA TGC GGG GTG GAG AAT GGT TGC CGC ACT TCG GCT TCA CTG CAG ACT TGA CGA AGC TT/3ThioMC3-D/-3′) and Neuropeptide Y (NPY) aptamer (5′-/5AmMC6T/GA ATG GAT CCA CAT CCA TGG ATG GGC AAT GCG GGG TGG AGA ATG GTT GCC GCA CTT CGG CTT CAC TGC AGA CTT GAC GAA GCT GA CGA A/3ThioMC3-D/-3′) were purchased from IDT-DNA (Coralville, IA, USA). The sequences of aptamer variants are in Table A1 of Appendix A. Amino functionalized magnetic nanoparticles (15 nm), glutaraldehyde, NHS ester-methylene blue (MB-NHS), tris (hydroxymethyl) aminomethane (TRIS), 6-mercapto-1-hexanol (MCH), NaCl, and MgCl$_2$ were purchased from Sigma-Aldrich. We used buffer as follows unless specified: Tris 100 mM, NaCl 100 mM, KCl 100 mM, and MgCl$_2$ (pH 8.3). The microchip platform was microfabricated by standard photolithography methods to create patterned vapor-deposited electrodes on glass, with a 3 mm diameter fluidic chamber to isolate the busing lines from the buffer. Its electrochemical performance was cross-checked using screen-printed gold electrodes (auxiliary: Pt; reference: Ag) purchased from DropSens.

### 2.2. Assembly of Methylene-Blue-Modified Aptamers on Gold

The gold surfaces were electrochemically cleaned with 50 mM H$_2$SO$_4$ in the potential range of −0.7 to 1.2 V, with a scan rate of 0.1 V/s. Thiolated aptamer (final concentration 10 μM) was mixed with TCEP (1 mM in pH 7.4 tris buffer) and incubated for 1 h to reduce oxidized thiol groups. Then, 20 μL of aptamer (10 μM) was dropped on the gold electrode and incubated overnight at 4 °C, and subsequently washed with both DI water and ethanol. Electrodes were then incubated in 1 mM MCH solution for 1 h to mask the unreacted Au sites, followed by washing with both ethanol and DI water. This step was followed by activation of amine groups at the end of the probes by 2% glutaraldehyde solution for 4 h. Then, 20 μL of MB-NHS (25 μg/mL) was dropped on electrodes and incubated for

another 4 h with continuous shaking. The MB was conjugated to the end of the probes through succinimide ester coupling. This assembly was stored at 4 °C until future use. Before electrochemical measurements, different concentrations of cortisol/NPY (20 µL) in 100 mM Tris buffer or 50% human serum were incubated for 3 h with continuous shaking. To execute the signal amplification technique, 20 µL of NHS ester-methylene blue modified nanoparticle (NP-MB-NHS) conjugate was dropped on electrodes and incubated for another 4 h with continuous shaking.

### 2.3. NMR Spectra

The imino proton spectra were obtained at 600 MHz on a Tecmag NMR spectrometer. The DNA (sequence of aptamer variants in Table A1 of Appendix A) was prepared at 0.3 mM in a PBS buffer containing 10% $D_2O$. Water suppression was accomplished using excitation sculpting [20]. The spectra were acquired using a sweep width of 20 ppm, and 1024 acquisitions were averaged for the spectra. Spectra were acquired with a 1 to 1.1 ratio of aptamer to cortisol at temperatures between 10 and 85 °C.

### 2.4. Nanoparticle Conjugation

Magnetic nanoparticles (NPs) (~15 nm) were used to mediate the redox signal. The assay of the target was performed first in 100 mM Tris buffer and then in 50% human serum to simulate a complex sample, with different sets of redox labels explored for each case. For detection in 100 mM Tris buffer, 30 µL of magnetic NP sample and 45 µL MB-NHS ester solution (1 mg/mL in DMF) were rotated at 200 rpm for 8 h at room temperature in dark. The reaction mixture was then centrifuged at 16,000 rpm for 45 min, and the supernatant was removed. For washing away excess MB-NHS, 1 mL DI water was added to the solid conjugate in the vial, mixed well, and centrifuged at 16,000 rpm for 45 min, and the supernatant was discarded. This step was repeated 4 times. The MB-NHS attached to magnetic NPs were obtained at the bottom. Following this, 300 µL of $NaCO_3$ solution (pH 8.3) was added to the bioconjugate precipitate to form a homogeneous dispersion and stored in a refrigerator at 4 °C until further use. For 50% human serum, the procedure was somewhat similar, but the concentration of magnetic NPs and MB-NHS was increased threefold, i.e., 90 µL of magnetic NPs and 135 µL MB-NHS ester solution (1 mg/mL in DMF) was initially rotated in a dark room. After following the above procedure, the final volume was made to 300 µL of $NaCO_3$ solution (pH 8.3) and stored in a refrigerator at 4 °C until further use. The latter method is expected to amplify the signal levels.

### 2.5. Instrumentation

Differential pulse voltammetry (DPV) was performed with a CHI 660D Electrochemical Analyzer (CH Instruments, Austin, TX, USA), by scanning from −0.6 to 0 V (vs. Ag/AgCl) with an increment of 0.004 V, an amplitude of 0.3 mV, a pulse width of 0.05 s, a sample width of 0.0167, and a pulse period of 0.1 s.

### 2.6. Validation Using Human Samples

All recruitment and data collection procedures were approved by the Air Force Research Laboratory Institutional Review Board (FWR20160031H) prior to the initiation of the experiments. Blood draws were performed at baseline and at strategic points during training exercises but were collected at the end of the training day. For each sample, a max of 30 mL of blood was collected into a serum separator tube (SST) and allowed to clot for 30 min at room temperature. Tubes were centrifuged for 15 min at $1000 \times g$. Serum was removed, separated into 200 µL aliquots, and stored at −80 °C. Serum sample biomarkers levels were determined using commercially available assays: Neuropeptide Y (Human NPY) ELISA kit (Catalog No. EZHNPY-25K; EMD Millipore, St. Charles, MO, USA), the sensitivity of the assay was 2 pg/mL, and cortisol RIA (Catalog No. 07-221105; MP Biomedicals, Solon, OH, USA). Assays were used following manufacturers' instructions.

Individual samples for sensor performance testing were chosen to reflect the range of cortisol and NPY values observed in different individuals.

## 3. Results

### 3.1. Aptamer Conformation after Cortisol Binding

NMR spectroscopy is a powerful tool to probe the structure of aptamers and the conformational alterations that accompany ligand binding. While the spectra for the DNA bases and sugars are often complicated by the overlap of their many similar resonances, the imino proton spectrum (10–15 ppm) is well resolved from the other signals and typically contains only a single peak for each base pair [9]. The imino protons exchange rapidly with water and can only be observed in the NMR spectra if they are protected from the rapid solvent exchange by the formation of stems and loops. The AT and GC base pairs appear in the spectral ranges of 12.5–14 ppm and 11.5–12.5 ppm, while the imino protons in protected loops and non-canonical base pairs typically appear between 9.5 and 11.5 ppm. The imino proton chemical shift depends, not on the type of base pair alone, but also on the local structure and binding of target ligands. The previously reported cortisol aptamer named A15-1 [10] (sequence of aptamer variants in Table A1 of Appendix A) has been used in different sensor platforms, but we sought to characterize the conformational alterations using NMR, to enable improvements in sensor design.

The mfold-predicted structure in Figure 1A shows three stem-loops [20], two of which involved the primer sequences used for amplification during the selection process. NMR spectra were used to assess structural features of this aptamer sequence and identify the cortisol-binding site in the A15-1 aptamer, by considering two truncations. A15-1-PF is the primer-free version of A15-1 created by cuts shown by the red lines in Figure 1A. Since the mfold analysis of the A15-1-PF aptamer [21] showed the stem-loop structure from the original analysis, as the only structured nucleotides, we created a further truncated aptamer (A15-1-SL) from this portion of the aptamer shown by the green lines in Figure 1A. Figure 1B shows the imino proton spectra for the 15-1 aptamer at 25 °C. In addition to the expected signals from the stem region for the AT and GC base pairs between 12.7 and 13.2 ppm, broad signals are observed in the region between 10 and 12 ppm. Figure 1C shows the imino proton spectra for the aptamer as a function of temperature between 10 and 85 °C. As expected, the stem-loop structure melted with increasing temperature, and a loss in signal intensity in the imino proton in the AT and GC regions by ~35 °C was observed, similar to the behavior reported for the cocaine-binding aptamer [22–24]. The data show the disappearance of the GC/AT peaks by 35 °C, but the signals from the peaks between 10 and 12 ppm persist far above the expected melting temperature of a stem-loop structure. Taken together, these NMR data suggest that multiple conformations may coexist in the A15-1 aptamer under our experimental conditions. Furthermore, only peaks in the AT and GC regions (12–14 ppm) are expected from the mfold structure. The NMR data suggest that the stem-loop portion of the A15-1 aptamer could form a G-quartet (G4) structure [25]. G4s are known to produce signals in the range of 10–12 ppm, which are more stable than duplex DNA [26]. Consistent with the formation of the G4, the NMR data in Figure 1B shows that the exchangeable protons in the range of 10–12 ppm can be observed at temperatures as high as 85 °C, while the AT and GC protons are lost from the spectrum by 45 °C. These data affirm that the stem-loop portion of the A15-1 aptamer forms a G4. This result is reasonable, given the high G content of the stem-loop, as shown by the red circles in Figure 1A. The formation of G4 is also consistent with a score of 20 from the GCRS mapper web server [27] that is used to predict G4 formation. This score is similar to the well-known G4 thrombin-binding aptamer [27] which has a GCRS score of 21. The broad lines in the range of 10–12 ppm also suggest multiple possible G4 conformations.

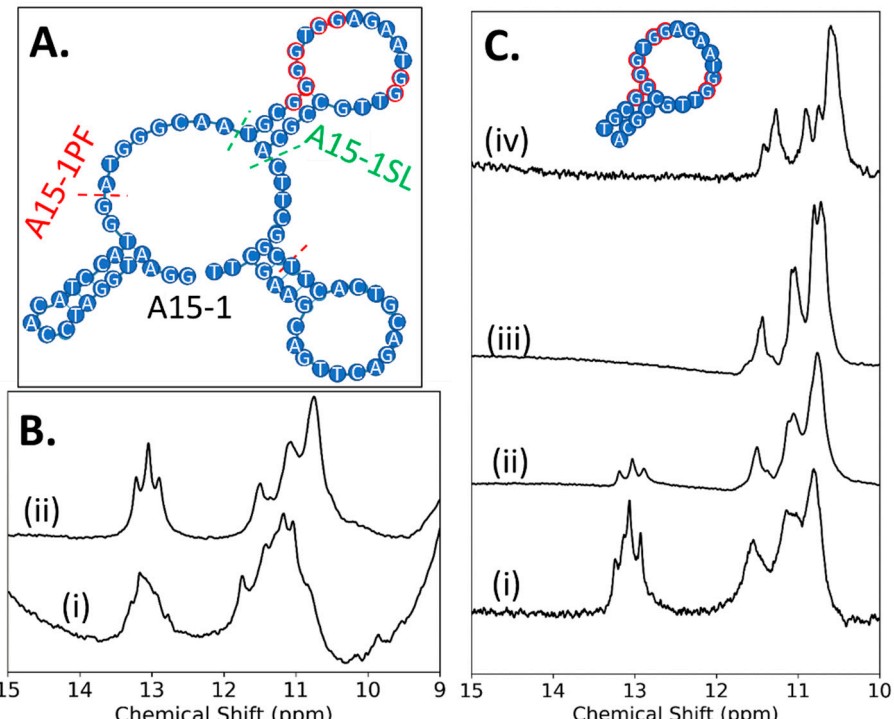

**Figure 1.** (**A**) The A15-1 aptamer folded using the mfold2 web server [21]. Primer sequences are shown with small letters and the G nucleotides that potentially form G-quartets are shown in red. The 400 MHz proton NMR spectrum of the exchangeable protons of (**B**) A15-1-PF (i) and A15-1-SL (ii) at 25 °C; (**C**) the NMR spectra of A15-1-SL at 10 (i), 25 (ii), 45 (iii), and 85 °C (iv).

Figure 2A shows the imino proton spectra for the A15-1 aptamer in the absence and presence of cortisol for the 1:1 complex at a concentration of 0.3 mM. No significant changes are observed in the imino proton spectra at 25 °C with the addition of cortisol. Figure 2B shows the NMR spectrum between 0 and 2 ppm, which contains many DNA and cortisol signals. The cortisol C18 methyl protons appear at the highest field (0.78 ppm) and are shown by the dotted line. The spectra for A15-1PF and A15-1SL aptamers in the presence of cortisol show that the C18 methyl protons are shifted upfield and broadened. Taken together, these data suggest that cortisol binds the A15-1 aptamer but does not lead to a large change in structure upon binding. Hence, derivatization of the 15-1 aptamer with magnetic nanoparticles is explored as a means to disrupt the DNA structure and promote a conformation change that will facilitate the generation of an electrochemical signal through a methylene blue (MB) signal tag.

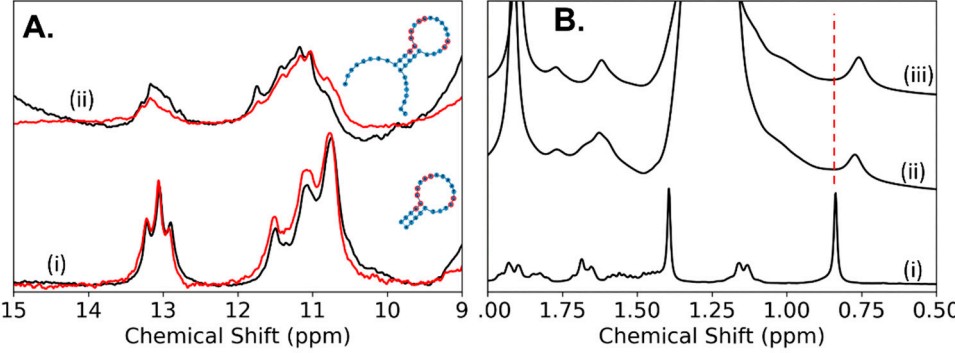

**Figure 2.** (**A**) The exchangeable proton spectrum of A15-1-sl (i) and A15-1-pf (ii) in the absence (black) and presence (red) of cortisol. (**B**) The upfield portion of the spectrum for cortisol (i) and the 1:1 cortisol complex for A15-1-sl (ii) and A15-1-pf (iii).

### 3.2. Nanoparticle-Mediated Signaling for Sensor Development

The two modalities of the sensor within this work are shown in Scheme 1A,B. Per Scheme 1A, attachment of a single methylene blue (MB) species to the aptamer creates a signal OFF assay, i.e., the turning OFF of MB signaling after aptamer binding to target as per Scheme 1A(i) vs. Scheme 1A(ii). In the second modality (Scheme 1B), nanoparticles conjugated with multiple MB signal moieties were attached to the aptamer to create a signal ON assay. Herein, MB signaling is initially absent due to the repulsion of nanoparticles from the gold surfaces, which is attributed to their high negative zeta potential, which resembles the surface charge on the gold surface. However, upon specific binding to the target molecule, the aptamer conformation change in the aptamer (Scheme 1B(i) vs. Scheme 1B(ii)) likely forces the MB species to be placed in proximity to the gold surface (explained further in Scheme 2), thereby turning ON the MB signal. This was applied to the detection of NPY and cortisol (C) on the microchip platform (Scheme 1C). Figure 3A shows voltammograms (DPV) confirms that signaling can be attributed to MB on the aptamer, based on the lack of signal from the indicated control samples. Upon cortisol binding by assays from Scheme 1, signal OFF (Figure 3B) or signal ON (Figure 3C) are created for target in the µg–pg/mL range, with the inset (i) showing a magnified view of the scans from Figure 3B.

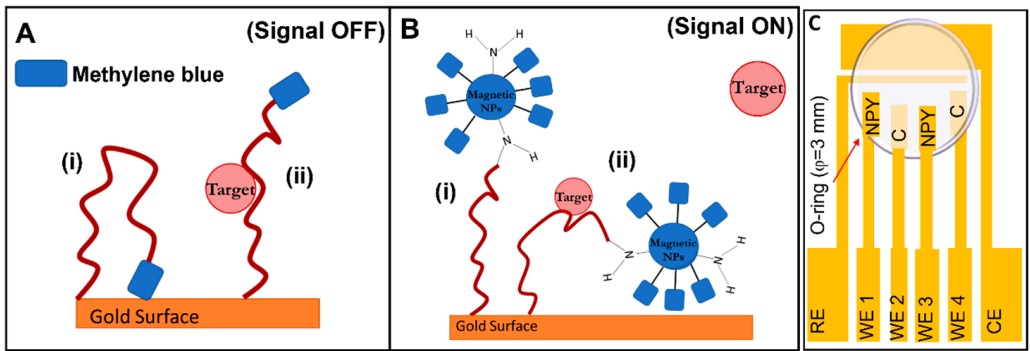

**Scheme 1.** Nanoparticle (NP) mediated methylene-blue signaling to cause signal OFF assay to signal ON assay with amplification: (**A**) signal OFF after (ii) versus before (i) target binding; (**B**) signal ON after (ii) versus before (i) target binding; (**C**) electrically addressable microchip with different aptamers (NPY or C for cortisol) immobilized on 500 µm working electrodes (WE) for signal transduction.

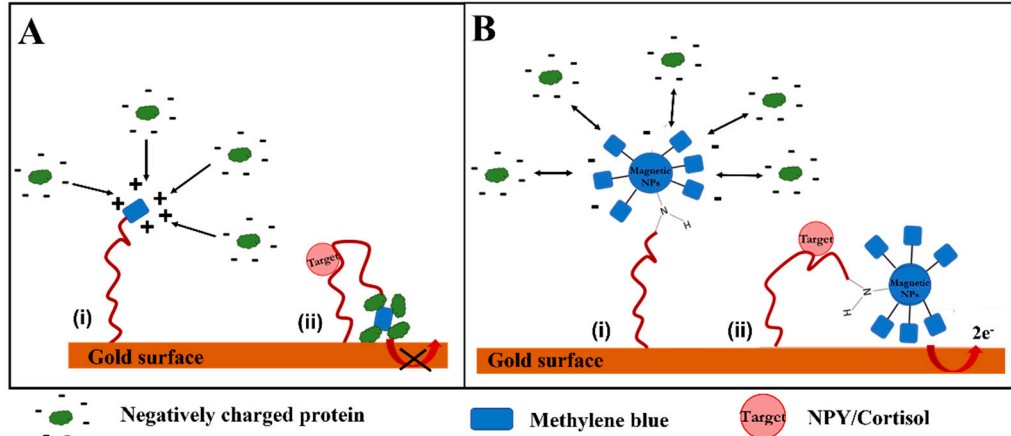

**Scheme 2.** MB signaling for single MB label (**A**) vs. for nanoparticle-immobilized MB (**B**).

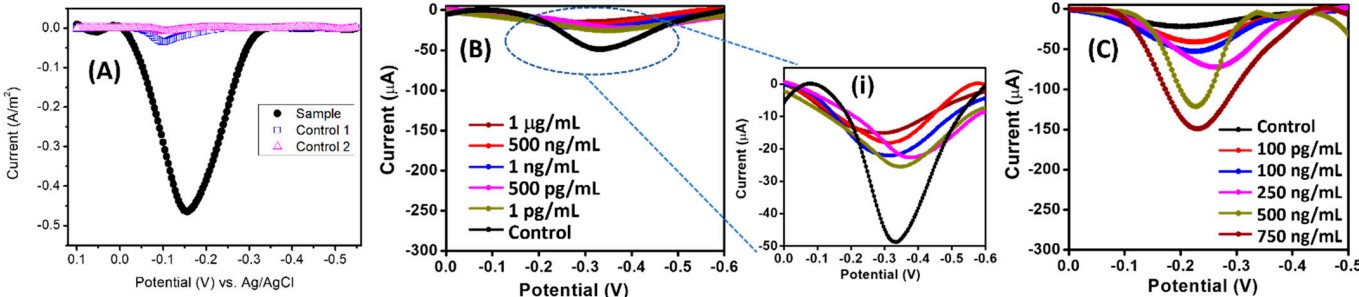

**Figure 3.** Differential pulse voltammograms (DPV): (**A**) MB signaling due to reaction of Au/NPY aptamer (5′ amine) with NHS-MB through amine-NHS crosslinking. Control 1: Au/MCH incubated in NHS-MB (no amine groups to react with NHS). Control 2: Au/NPY aptamer in MB (no NHS to react with amine on aptamer); (**B**) signal OFF and signal ON, expanded in inset (i) to show the signals; (**C**) sensing paradigms for cortisol detection in the μg–pg/mL range.

Calibration plots for cortisol and NPY measurements over the μg–pg/mL range are shown in Figure 4. Comparing (Figure 4A–D), the signal ON assay is clearly preferable over the signal OFF assay, due to its greater sensitivity (higher slope of the plots), the lower standard deviations at detection limit levels, and the need for lower aptamer concentrations, which would reduce costs. The nanoparticle conjugate with multiple MB sites to a single nanoparticle likely contributes to more sensitive measurements.

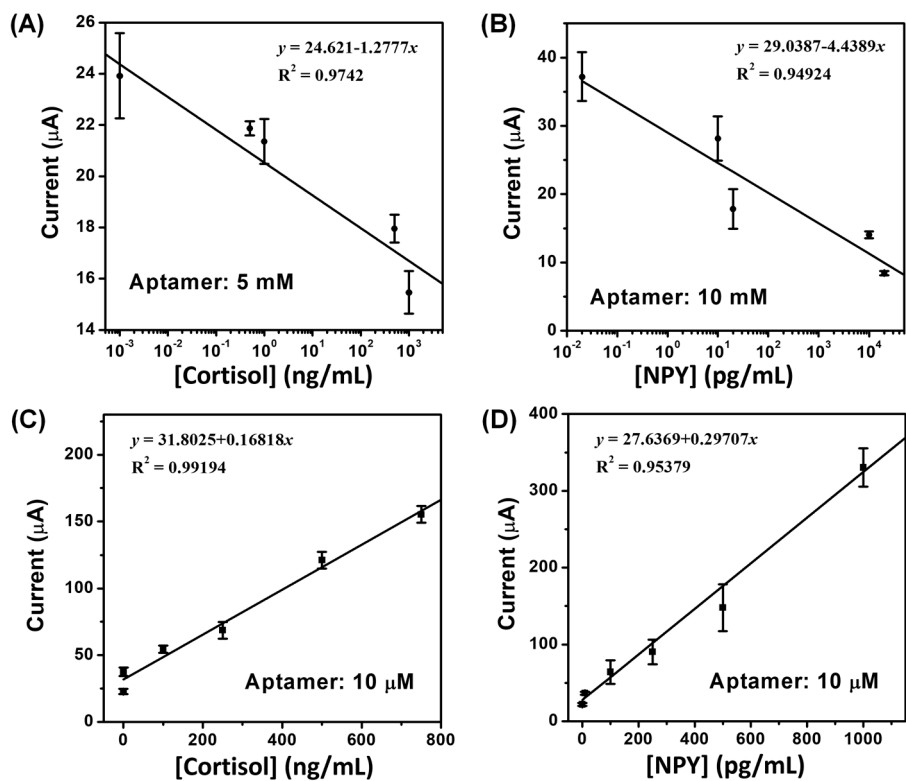

**Figure 4.** Calibration plots for signal OFF (**A**,**B**) vs. signal ON assay (**C**,**D**) of cortisol (**A**,**C**) and NPY (**B**,**D**) spiked into 100 mM Tris buffer.

### 3.3. Multitarget Sensing and Validation with Serum Samples

Based on the calibration plots in Figure 4, the operation of the sensor platform in serum media was explored by spiking cortisol in corticoid-free serum media after dilution by 50% with Tris buffer. As per the voltammograms in Figure 5A, the signal OFF assay of Scheme 1A is not able to distinguish the control sample with no cortisol versus the 1 μg/mL cortisol sample in serum media, whereas this distinction was easily accomplished in Tris

buffer (Figure 3B(i)). On the other hand, the signal ON assay of Scheme 2B can distinguish the control sample with no cortisol versus the sample spiked with 50% diluted cortisol in serum media at various levels.

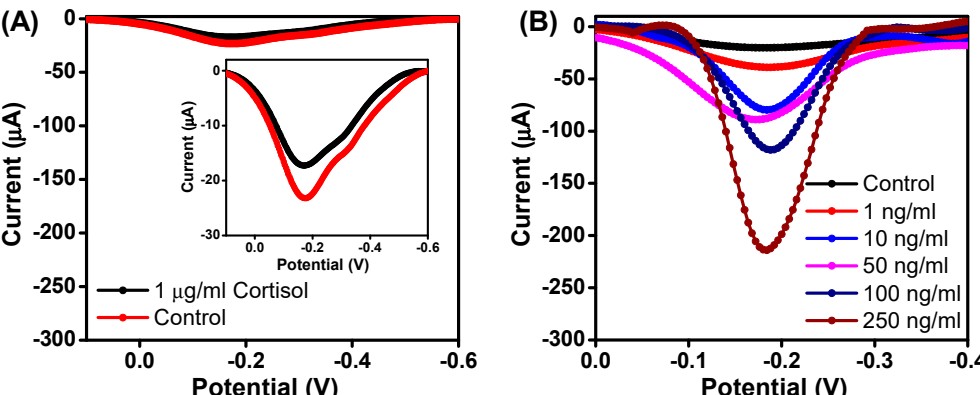

**Figure 5.** DPV measurements of cortisol in 50% diluted serum show: (**A**) shutdown of MB signaling for signal OFF assay; (**B**) maintenance of MB signaling for signal ON assay.

Based on similar investigated approaches [28], we suggest that this shut down of MB signaling in serum media is likely due to the strong interaction of positively charged MB with negatively charged serum proteins (Scheme 2A). On the other hand, conjugation of MB to the considerably larger nanoparticles (15 nm) causes their net charge type to resemble that of serum proteins (Scheme 2B(i)) since both species exhibit negative zeta potential at the pH used in the detection buffer. Hence, the interaction of MB with serum proteins is avoided, causing the signal ON assay (Scheme 2B(ii)).

Based on this, the signal ON assay was used to construct calibration plots of the respective analytes in the 50% serum media (Figure 6A,B). Typical NPY levels in serum samples vary from 2–1000 pg mL$^{-1}$ [29], while cortisol in serum samples taken in the morning range between 70 and 250 ng mL$^{-1}$, and those taken in the evening are in the range of 20–140 ng mL$^{-1}$ [30]. For healthy individuals under normal conditions, the NPY level in blood is between 0.6 and 2.6 pg/mL, and the cortisol is between 30 and 230 ng/mL [31], with the levels increasing under stress. Hence, it was encouraging to observe that the detection ranges for our sensor (Figure 6A,B) overlap with this working range for both NPY and cortisol in 50% diluted serum. The subjects (Sub 1 and Sub 2) were monitored over multiple days (D1, D2, and D3), shown in Figure 6C,D, as needed to assess stress-induced alterations. Our prior research [13] confirmed that testosterone and estradiol, when present at their highest level of the typical range in serum samples of 10 ng/mL and 400 pg/mL, respectively [32], did not interfere with the binding of this aptamer to the cortisol target, down to its lowest levels of 60 ng/mL in serum samples [33]. However, progesterone at its highest level within serum (1 μg/mL) can increase the signal from cortisol at its lowest level within serum (60 ng/mL) by about 15%, but the kinetics of aptamer binding to cortisol is relatively unaltered (especially the binding parameter: $k_{on}$).

Following this, multitarget measurements of cortisol and NPY were conducted using 50% diluted human subject serum samples for quantification by comparison versus the calibration plots. The results in Figure 6C,D indicate that the described sensor can sensitively and simultaneously detect the variations in NPY and cortisol levels within the samples. A summary of the sensitivity, LOD, LOQ, and linear concentration range for the targets based on OFF and ON signaling approaches is in Table A2 of Appendix A.

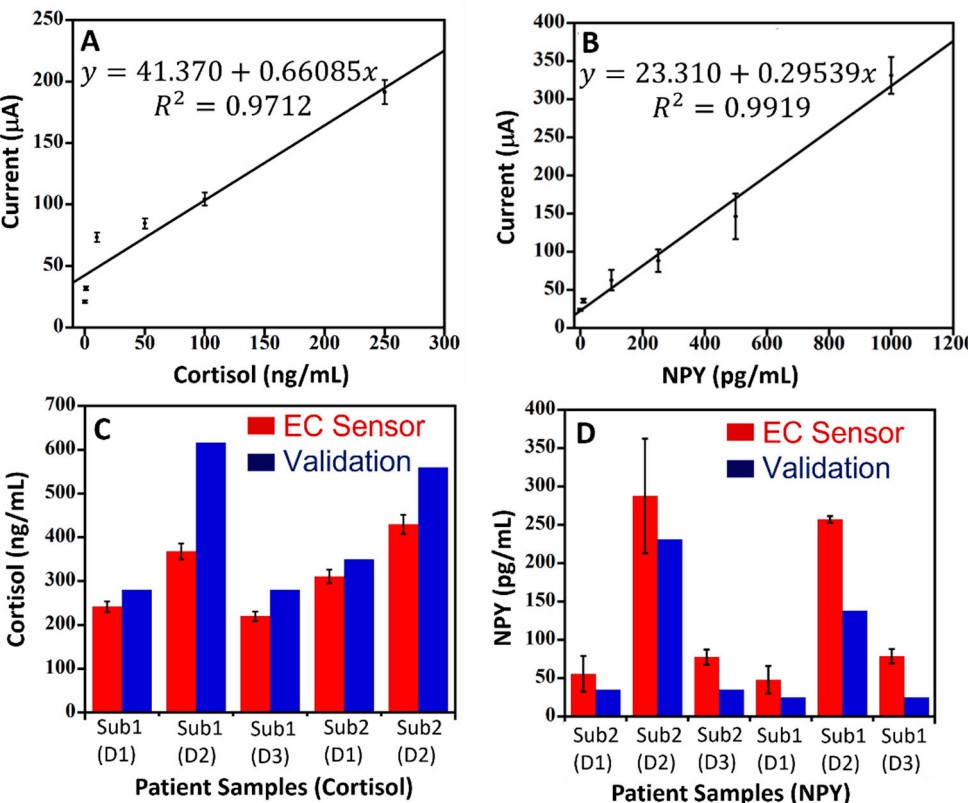

**Figure 6.** Calibration plots for cortisol (**A**) and NPY (**B**) spiked into 50% serum medium and validation of the current sensor versus immunoassay measurements from human serum samples for cortisol (**C**) and NPY (**D**) levels. Samples were from two subjects (Sub1 and Sub2) collected over three days (D1–D3).

## 4. Conclusions

Aptamer-based sensors for multiplexed detection of cortisol and NPY were developed over the physiologically relevant range (μg–pg/mL). Based on NMR spectra, we infer that the aptamer structure is not altered significantly upon binding, leading us to explore the derivatization of the aptamer with magnetic nanoparticles to promote a conformation change that will facilitate the generation of an electrochemical signal through a methylene blue (MB) signal tag. In absence of aptamer modification, a signal OFF assay is obtained, but quantification is limited in serum media, based on the inability to distinguish the control sample with no cortisol versus the 1 μg/mL cortisol sample in serum media. Hence, magnetic nanoparticle-mediated signal ON aptamer assays for cortisol and NPY were developed to distinguish the control sample with no cortisol versus the sample spiked with 50% diluted cortisol in serum media at various levels. We suggest that the shutdown of MB signaling in serum media is likely due to the strong interaction of positively charged MB with negatively charged serum proteins, whereas this interaction is avoided for the signal ON assay, due to the like charges that occur on the large nanoparticle to resemble those on serum proteins. A calibration plot for measurement of cortisol and NPY in 50% serum media was completed for its application towards multitarget measurements of cortisol and NPY using 50% diluted human subject serum samples. The results indicate that the described aptasensor can sensitively and simultaneously detect the variations in NPY and cortisol levels within the samples, as validated by its correspondence to measurements conducted by immunoassays.



**Author Contributions:** N.S.: Investigation for Sensing, Formal analysis, Data Curation; Y.L.: Methodology, Investigation for Sensing; R.S.: directed IRB submission and subject studies; P.M.: Investigation for NMR experiments and data analysis, J.L.C.: Conceptualization, Supervision; N.S.S.: Conceptualization, Methodology, Formal analysis, Resources, Writing, Supervision, Project administration, Funding acquisition. All authors have read and agreed to the published version of the manuscript.

**Funding:** This study was funded by US AFOSR contract FA2386-21-1-4070, Air Force Research Laboratory under agreement number FA8650-18-2-5402, and research contracts from UES, Inc. (UES subcontract S-168-1X5-001 to prime contract FA8650-19-D-6109/FA8650-19-F-6110) and Cambridge Medical Technologies (CMT), Inc.

**Institutional Review Board Statement:** All recruitment and data collection procedures were approved by the Air Force Research Laboratory Institutional Review Board (FWR20160031H) prior to the initiation of the experiments.

**Informed Consent Statement:** Informed consent was obtained from all subjects involved in the study.

**Data Availability Statement:** The data that support the findings of this study are available in the Appendix A and from the corresponding author upon reasonable request.

**Conflicts of Interest:** The authors declare no conflict of interest.

## Appendix A

**Table A1.** Sequences of the presented aptamer variants.

| Aptamer | Sequence |
| --- | --- |
| CSS3.51 | CTCTCGGGACGACGCCAGAAGTTTACGAGGATATGGTAACATAGTCGTCCC |
| CSS3.42 | GGACGACGCCAGAAGTTTACGAGGATATGGTAACATAGTCGT |
| CSS3.29 | GCCAGAAGTTTACGAGGATATGGTAACATA |
| A15-1 | GAA TGG ATC CAC ATC CAT GG ATG GGC AAT TGC GGG GTG GAG AAT GGT TGC CGC ACT TCG GGC TTC ACT GCA GAC TTG ACG AAG CTT |
| A15-1PF | ATG GGC AAT TGC GGG GTG GAG AAT GGT TGC CGC ACT TCG GGC |
| A15-1SL | TGC GGG GTG GAG AAT GGT TGC CGC |

**Table A2.** Sensor metrics for the targets using signal OFF and signal ON approaches.

| Sample Type | Signal | Sensitivity | LOD | LOQ | Linear Range |
| --- | --- | --- | --- | --- | --- |
| Cortisol in Buffer | OFF | 1.278 µA ng$^{-1}$ mL | 0.2 ng/mL | 0.6 ng/mL | 1 pg/mL–1 µg/mL |
| | ON | 0.1682 µA ng$^{-1}$ mL | 1.5 ng/mL | 4.5 ng/mL | 0.1–750 ng/mL |
| Cortisol in 50% Serum | ON | 0.6609 µA ng$^{-1}$ mL | 0.9 ng/mL | 2.7 ng/mL | 1–250 ng/mL |
| NPY in Buffer | OFF | 4.439 µA pg$^{-1}$ mL | 0.1 pg/mL | 0.2 pg/mL | 0.2 pg/mL–2 ng/mL |
| | ON | 0.2971 µA pg$^{-1}$ mL | 2.2 pg/mL | 6.7 pg/mL | 1–1000 pg/mL |
| NPY in 50% Serum | ON | 0.2954 µA pg$^{-1}$ mL | 2.0 pg/mL | 5.9 pg/mL | 1–1000 pg/mL |

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
