# Peer review of "Nanoparticle-Mediated Signaling for Aptamer-Based Multiplexed Detection of Cortisol and Neuropeptide Y in Serum"

_chemosensors, doi:10.3390/chemosensors10050153_

Round 1

Reviewer 1 Report

The manuscript „Nanoparticle-Mediated Signaling for Aptamer-Based Multiplexed Detection of Cortisol and Neuropeptide Y in Serum” by Sardesai et al describes the development of an biosensor which is able to detect Cortisol and Neuropeptide Y at the same time. The authors developed different sensors using different assays, a signal OFF and a signal ON sensor. Nevertheless for the detection in serum only the signal ON assay is sensitive enough to detect cortisol and Neuropeptide Y in serum. With there article the authors address a big problem in the field of biosensors, which is how to detect multiple molecules in one assay. Therefore the article is well in the scope of “chemosensors”.

The article is well written and the reader is able to follow the ideas and conclusions of the authors pretty well. 

Therefore I recommend to publish the article in “chemosensors” with only minor changes  

  1. In Fig. 1A the authors show the structure of the aptamer. If the article is printed it is not possible to see the letters for the nucleotides. The authors should enlarge the letters for better understanding.
  2. In Appendix table 1 the sequences of the different aptamers are shown. Unfortunately the CSS3 aptamer is not mentioned in the text. If it is not needed the authors should exclude it from the table.
  3. In line 170 the “the” should be deleted

Author Response

Please see attached pdf file

Reviewer 2 Report

The authors described an electrochemical sensor model for multiplex detection of NPY and cortisone on different channels.

NMR studies were conducted to show the structural differences of the target-specific aptamers upon target binding in which  aptamer derivates were tested.

However, I cannot see the correlation between the NMR study and the sensor design. Overall, the NMR studies seem to not contribute to the designated work. Instead, the authors claim the structural change by bead conjugation, for which there is not any experimental evidence.

Maybe testing of several other aptamer derivates could have worked better to find the one that can create a desirable shift for sensing. 

Lastly, I cannot see the evidence for aptamer binding to the surface or the dye binding to the aptamers on the electrode surface.

Reviewer 3 Report

Minor revision

This manuscript can be accepted for Chemosensor after additions and corrections.

  1. I recommend to add physiological level of neuropeptide Y (NPY) and cortisol in serum of healthy donor and compare this level with level as stress response.

  1. Authors envision the application of this sensor in a wearable format, however the description of such format must be added in more details.
  2. More detailed description of commercially available ELISA kits for serum sample biomarkers levels (NPY) and RIA (cortisol) is necessary.
  3. In Material and Method section (2.5) additional information concerning parameters of DPV is desirable.
  4. Authors did not use cyclic voltammetry in this paper. This method must be deleted from section 2.5.
  5. I suppose that Figure 3 and 5 represents differential pulse voltammetry experiments. Figure 3 and 5 needs addition on figure legend.
  6. Methylthioninium chloride, commonly called methylene blue, it is a salt, authors must explain why they think that MB binds with positively charges proteins from serum.
  7. Comparison of sensitivity, LOD, LOQ, linear concentration range of neuropeptide Y (NPY) and cortisol for two approaches (such as OFF and ON) must be added in discussion.

Author Response

We thank the reviewer for careful examination of the manuscript and contributed review comments. Please see attached revised manuscript with highlighted changes.

(1) I recommend to add physiological level of neuropeptide Y (NPY) and cortisol in serum of healthy donor and compare this level with level as stress response.

Response: We had this information in the original manuscript (references [29] and [30]), but we have added another citation ([31]) and some explanation.

Alterations to the manuscript (line 273-275): For healthy individuals under normal conditions, the NPY level in blood is between 0.6 and 2.6 pg/ml, and the cortisol is between 30 and 230 ng/ml [31], with the levels in-creasing under stress.

(2) Authors envision the application of this sensor in a wearable format, however the description of such format must be added in more details.

Response: As requested, we have rephrased a line in the Introduction section to explain this vision

Alterations to the manuscript (line 66): “We envision the application of this sensor in a wearable format [18], to extract markers from interstitial fluid for profiling a wider array of human performance biomarkers from biological matrices [19], under stress-related events to develop assessment and augmentation methodologies.”

(3) More detailed description of commercially available ELISA kits for serum sample biomarkers levels (NPY) and RIA (cortisol) is necessary.

Response: As requested, we have added this information to the Methods section under “Validation using human samples” subsection.

Alterations to the manuscript (line 137-141): “Serum sample biomarkers levels were determined using commercially available assays: Neuropeptide Y (Human NPY) ELISA kit (Catalogue No. EZHNPY-25K; EMD Millipore, St. Charles, MO), sensitivity of the assay was 2 pg/mL., and cortisol RIA (Catalogue No. 07-221105; MP Biomedicals, Solon, OH). Assays were used following manufacturers’ instructions.”

(4) In Material & Method section (2.5) additional information concerning parameters of DPV is desirable.

Response: As requested, we have added this information to the Methods section under the “Instrumentation” subsection.

Alterations to the manuscript (line 127-129): “Differential pulse voltammetry (DPV) was performed with a CHI 660D Electrochemical Analyzer (CH Instruments, Austin, TX, USA), by scanning from -0.6 to 0 V (vs. Ag/AgCl) with an increment of 0.004 V, an amplitude of 0.3 mV, a pulse width of 0.05 s, a sample width of 0.0167, and a pulse period of 0.1 s.”

(5) Authors did not use cyclic voltammetry in this paper. This method must be deleted from section 2.5.

Response: As requested, we have deleted it.

(6) I suppose that Figure 3 and 5 represents differential pulse voltammetry experiments. Figure 3 and 5 needs addition on figure legend.

Response: As suggested we modified the figure legends of Fig. 3 and Fig. 5.

Alterations to the manuscript: We have changed the respective captions as follows:

Line 236: Figure 3. Differential pulse voltammograms (DPV) show signal OFF (A) and signal ON (B) sensing paradigms for cortisol detection in the µg-pg/ml range.

Line 258. Figure 5. DPV measurements of cortisol in 50 % diluted serum show: (A) shutdown of MB signaling for signal OFF assay; (B) maintenance of MB signaling for signal ON assay.

(7) "Methylthioninium chloride, commonly called methylene blue, it is a salt, authors must explain why they think that MB binds with positively charges proteins from serum."

Response: The reviewer may have misunderstood how we use methylene blue (MB) in this work.  We did not use free MB in solution, rather the MB was covalently attached on the aptamer. When attached to the nanoparticle per Scheme 2B, we suggest that the high negative zeta potential of the nanoparticle repulses non-specific binding of serum proteins to the aptamer.

(8) Comparison of sensitivity, LOD, LOQ, linear concentration range of neuropeptide Y (NPY) and cortisol for two approaches (such as OFF and ON) must be added in discussion.

Response: As suggested, we add a table (Table 2 of Appendix) that summarizes the information.  LOD and LOQ are obtained using the following equations:

LOD = 3.3 σ/S, and LOQ = 10 σ/S 

Here, σ is the standard deviation of the data between -0.3 and 0 V of the blank (i.e., Au/aptamer chips without MB), and S is the slope of the calibration curve.

Alterations to the manuscript (line 323): Table 2 of the Appendix section

Round 2

Reviewer 2 Report

I believe authors should show the confirmation experiments of the Au- aptamer-MB binding either in the manuscript or in the supplementary data files. 

Author Response

As requested by the reviewer, we have added this data as Figure 3A in the attached revision of the manuscript, with more detailed explanation of the controls in the caption.